# An Approach of Feed-Forward Neural Network Throughput-Optimized Implementation in FPGA

**Rihards Novickis ***[ID]**, Daniels Jānis Justs** [ID]**, Kaspars Ozols** [ID] **and Modris Greitāns** [ID]

Institute of Electronics and Computer Science, 14 Dzerbenes St., LV-1006 Riga, Latvia;
daniels.justs@edi.lv (D.J.J.); kaspars.ozols@edi.lv (K.O.); modris_greitans@edi.lv (M.G.)
***** Correspondence: rihards.novickis@edi.lv; Tel.: +371-67558-288

**Abstract:** Artificial Neural Networks (ANNs) have become an accepted approach for a wide range of challenges. Meanwhile, the advancement of chip manufacturing processes is approaching saturation which calls for new computing solutions. This work presents a novel approach of an FPGA-based accelerator development for fully connected feed-forward neural networks (FFNNs). A specialized tool was developed to facilitate different implementations, which splits FFNN into elementary layers, allocates computational resources and generates high-level C++ description for high-level synthesis (HLS) tools. Various topologies are implemented and benchmarked, and a comparison with related work is provided. The proposed methodology is applied for the implementation of high-throughput virtual sensor.

**Keywords:** FPGA; feed-forward neural networks; throughput-optimization; virtual sensors; HLS

## 1. Introduction

Since ImageNet Image Classification competition was assuredly won by Krizhevsky, Sutskever and Hinton with their deep-learning-based solution in 2012 [1], it became evident that Deep Learning (DL) algorithms bear the potential for a variety of applications. With the increasing availability of computational resources, Machine Learning (ML) has become a widely used technique for solving a variety of different problems, e.g., object identification, cluster classification, pattern recognition, functional regression, etc. [2]. Ever since Deep Neural Networks (DNNs) demonstrated their superior performance, they have been considered for a wide range of use-cases and processing architectures. Considerable effort has been devoted to improving computational efficiency by developing new Artificial Neural Network (ANN) architectures [3,4] and optimizing implementations for specific use-cases [5–7].

DNNs constitute their power through massively parallel distributed structures and their ability to learn and, therefore, generalize [8]. These two information processing capabilities raise the potential of solving complex problems. The parallel nature suggests that it is worthwhile to implement DNNs in parallel architectures, e.g., Field Programmable Gate Arrays (FPGAs).

FPGA technology was developed in the middle of 1980s with the original intent to be a prototyping medium [9]. As the underlying silicon technology improved, it became widely accepted in most communication technologies. Currently, FPGAs are well established in almost all areas of computing. The potential of ML has been widely recognized by major FPGA vendors [10–13] and has even led to the creation of specialized development flows [14]. Nevertheless, resource availability and costs limit FPGA usage in ML applications. A notable effort has been devoted to solving this problem by designing different hardware architectures [15].

In this article, authors examine Feed-Forward Neural Networks (FFNNs) and investigate their implementation by adopting a pipelining design technique. It is proposed to revise the implementation

challenge and view the problem in terms of elementary structures. FFNN is split into elementary layers, where each layer can be characterized by its resource and its respective latency, e.g., adder, multiplier, activation function. The resource count is varied between layers so that the latency throughout the network is evenly distributed, thus resulting in an optimally pipelined implementation. A dedicated tool has been designed to convert the given network's topology into C++ code which is compatible with High-Level Synthesis (HLS) tools. The generated code incorporates the necessary directives to ensure a resource-limited and pipelined solution. The created tool is made available online (http://git.edi.lv/rihards.novickis/generation_tool_hls_c_fully_connected_feed_forward_neural_network) and is used for different, previously published topologies, for which the results are compared. Finally, the developed methodology is validated by implementing a virtual sensor for a torque vectoring use-case.

The rest of this paper is organized as follows. Section 2 is devoted to the related work in the field, Section 3 describes the theoretical background of FPGAs and FFNNs, Section 4 describes different FFNN FPGA design considerations which are relevant for the approach, the proposed approach itself is described in Section 5, FFNN implementation and comparison with related work is discussed in Section 6, Section 7 describes the adoption of the design methodology for a virtual sensor use-case and concluding discussion is carried out in Section 8.

## 2. Related Work

Ever since early formalization of NNs [16], a significant effort has been made and various paradigms used to adopt different NN structures for digital circuit implementation. For example, Convolutional Neural Networks (CNNs) are widely used for image recognition and classification, although classification itself is carried out by a fully connected FFNN. Different kinds of paradigms ranging from co-processor systems [17] to OpenCL-based solutions [18–22] have been used to find the optimal trade-off between resource use, latency, and throughput. This section highlights relevant aspects of previous FFNN implementations and summarizes the main design challenges.

An implementation of a FFNN using VHDL is presented in [23]. The proposed network is based on a collection of simple-interconnected processing elements (neurons), which are organized into a topology composed of individual layers. The neurons between layers can communicate concurrently. Network's coefficients are represented using one's complement signed fixed-point binary numbers. Different hardware optimization techniques have been suggested, i.e., weight storage in internal memory, Booth's multiplication algorithm and activation function's bilinear approximation using a counter and shift registers. The authors implemented a simple FFNN with a 2-2-1 topology. Network's pipelined performance is estimated to be 34 ns per single output estimation, although latency and clock period is not provided.

Joint software and hardware implementation is presented in [24]. Architecture is based on a control unit, neurons and shared Look-Up Table or LUT-based activation function. The implementation's control unit uses user-defined code to dynamically load weights and inputs, store neuron outputs and reset accumulators in neuron cells. The paper investigates two simple topologies with 1 and 4 neurons. The implementations are tested against time series prediction network, which uses a 2-4-1 structure. The provided solution's maximum performance is 0.66 µs and 0.44 µs for 1 and 4 neuron implementations respectively.

It is important to achieve reduced area and increased performance of the circuit, but this becomes increasingly difficult to carry out if low approximation error is required. In Refference [25], the authors propose a hybrid approximation method of hyperbolic tangent activation function, which takes into account the linear nature of the hyperbolic tangent when the argument value is small. This approach is combined with a bit-level mapping of function's non-linear region. Bit-level mapping returns an average value of a sub-range of the region being approximated. Sub-ranges are split so that the approximation error is below a certain threshold.

The approach [25] is used in an optical character recognition system [26], where a FFNN is embedded into an FPGA. The authors use the aforementioned hybrid approximation method to

implement a hyperbolic tangent activation function. The selected network's topology is 189-160-36, and implementation's processing time is 4.36 µs.

A reconfigurable neural network architecture, composed of 20 neurons, is proposed in [27]. Architecture is divided into four parts: instructions unit, memory unit, layer unit and controller unit. Architecture adopts 8-bit precision. The approximation of the activation function is based on direct transformation from input to output. The network is tested using 4-8-3-3 and 1-5-1 topologies. The architecture is generic, i.e., it applies to different topologies without reconfiguration. The implementation uses VEDIC multiplier instead of available on-chip Digital Signal Processing (DSP) blocks.

A valuable work exploring floating-point based implementation is carried out in [28]. Two approaches-resource-saving and parallel-have been developed. The exponent, used for hyperbolic tangent and sigmoid function, is calculated using Padé approximation. The approach is benchmarked with 5-16-12-16-5 topology and Xilinx ZEDBoard evaluation board with Zynq XC7020 chip. The authors illustrate that implementation is advantageous over high-performance software platforms due to its parallel execution.

In conclusion, the intrinsic programmable logic's parallel nature suggests its suitability for the implementation of FFNNs. Although different architectural approaches and design choices have been investigated, FFNN implementations face a major challenge of limited hardware resources. Furthermore, expansion of the NN topologies [29] and saturation of the manufacturing process improvement [30] suggest the persistence of the resource challenge. The summary of different topologies and performance metrics of previous FFNN design approaches is provided in Table 1.

**Table 1.** Summary of the NN topologies and the performance metrics from the related articles.

| Paper | Data Type | Topology | Activation Function | Approximation Method | Latency (µs) | Throughput (Samples/s) |
|-------|-----------|----------|---------------------|----------------------|--------------|------------------------|
| [23] | Fixed $< 16, 7 >$ | 2-2-1 | Sigmoid/ Linear | Piece-wise linear | 0.034 | 29,412,000 (theoretical) |
| [24] | Fixed $< 32, 7 >$ | 2-4-1 | Sigmoid | LUT-based | 0.44 | 2,272,700 |
| [26] | Fixed | 189-160-36 | Hyperbolic tangent | Hybrid linear + bit-level mapping | 4.36 | 229,360 |
| [27] | Fixed $< 8, 3 >$ | 4-8-3-3 | Hyperbolic tangent /linear | Direct mapping | 1.16 | 862,070 |
| [27] | Fixed $< 8, 3 >$ | 1-5-1 | Hyperbolic tangent /linear | Direct mapping | 0.683 | 1,463,100 |
| [28]A | Single precision | 5-16-12-16-5 | Hyperbolic tangent /linear | Padé | 33,100 | 30.2 |
| [28]B | Single precision | 5-16-12-16-5 | Hyperbolic tangent /linear | Padé | 24,700 | 40.5 |
| [28]C | Single precision | 5-16-12-16-5 | Hyperbolic tangent /linear | Padé | 5700 | 175.4 |
| [28]D | Single precision | 5-16-12-16-5 | Hyperbolic tangent /linear | Padé | 3500 | 285.7 |

An interesting use-case in terms of potential application of our developed approach is presented in [31], where FFNNs enhance vehicle dynamics for a multi-motor electric vehicle. The authors train a predictive NN for the estimation of the future slip values of each wheel for a batch of possible torque-vectoring set points. These predictions determine the torque distribution that will reduce the unnecessary slip. Furthermore, authors benchmark the trained topologies using parallel computing platforms. One of the trained topologies is used in this article to validate the developed approach in a virtual sensor use-case, which will be further described in Section 7.

## 3. Background

Further on, the authors assume that the reader is familiar with FPGAs, pipelining principles of digital circuit design, HLS and basic theory of NNs. Nevertheless, as these concepts are essential part of the developed approach for the FFNN implementation, this section is devoted to describing them.

### 3.1. FPGA and Circuit Design

A simplified FPGA structure is illustrated in Figure 1. It consists of general logic, memory, DSP blocks, routing fabric and programmable input/output (I/O) blocks [32].

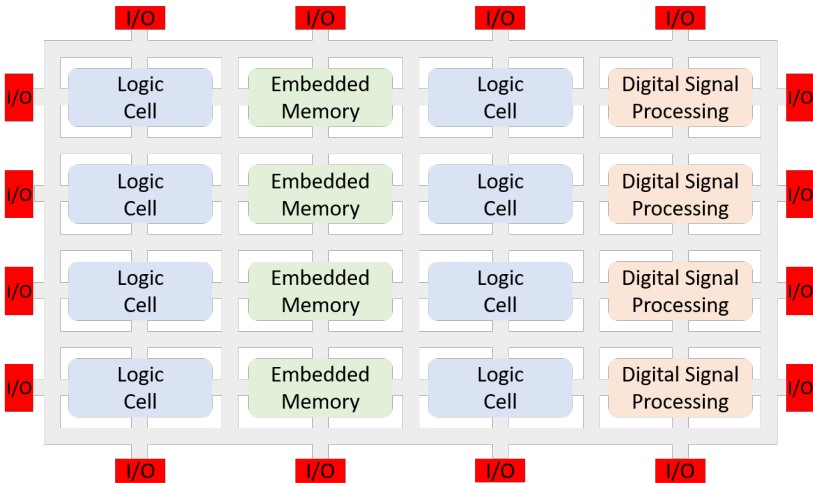

**Figure 1.** Superficial FPGA structure.

FPGA development often involves balancing between the performance and the resource use of the design. Unlike processor systems, where the execution of an algorithm is directed by the software, in FPGAs different physical parts of a chip can be dedicated to a specific task. These parts can be interconnected, enabling a special kind of concurrency-pipeline. Pipelining is an important technique to increase the system's performance by overlapping the processing of several tasks [33]. A fully pipelined solution can accept new input data on every clock cycle, and it is characterized by a fixed latency. The term Initiation Interval (II) denotes the number of clock cycles between consecutive input transactions. Conceptual examples of non-optimized and optimized processing pipelines are shown in Figure 2a,b, respectively. In these figures, functions represent different hardware blocks and DX denotes data samples.

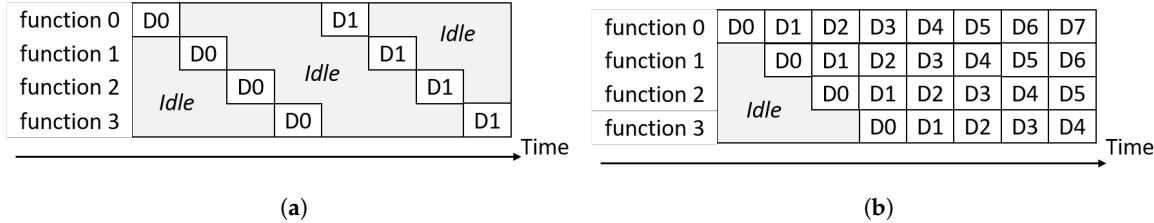

(**a**)                                                  (**b**)

**Figure 2.** Comparison of optimized and non-optimized pipelines. (**a**) Non-optimized processing pipeline; (**b**) Optimized processing pipeline.

Hardware Description Languages (HDLs) allow designers to adapt existing tools for logic synthesis and to explicitly describe hardware. Unfortunately, this approach requires the designer to specify functionality at a low level of abstraction, where the cycle-by-cycle behaviour is completely specified [34]. The use of such languages requires advanced hardware expertise and involves cumbersome development which leads to increased time-to-market expenses.

To demonstrate the design principles, we have chosen an increasingly popular design methodology involving the use of High-Level Synthesis (HLS) [34]. These tools enable the designer to produce the circuit's functional description using high-level software languages, e.g., C, C++ and SystemC. In return, HLS tools generate a Register Transfer Level (RTL) description. While this approach still requires some hardware expertise, it improves the maintainability and enables rapid exploration

of the design space [34]. Nevertheless, it must be noted that HLS tools introduce additional overhead as a result of the introduced data path scheduling logic.

*3.2. Feed-Forward Neural Networks*

A *Neural Network* is a system that is designed to model an oversimplification of how the brain performs a particular task or a function of interest [8]. A network can be split into fundamental information-processing units-neurons, which form the basis for designing artificial neural networks. The block diagram in Figure 3 shows neuron's mathematical model. Neuron's inputs $x_i$ are multiplied by coefficients $w_{ki}$, referred to as weights, and summed up together with a bias $b_k$. This sum is passed to an activation function $\varphi$, which is used to normalize neuron's output, $k$ and $i$ designate the specific neuron and its corresponding input.

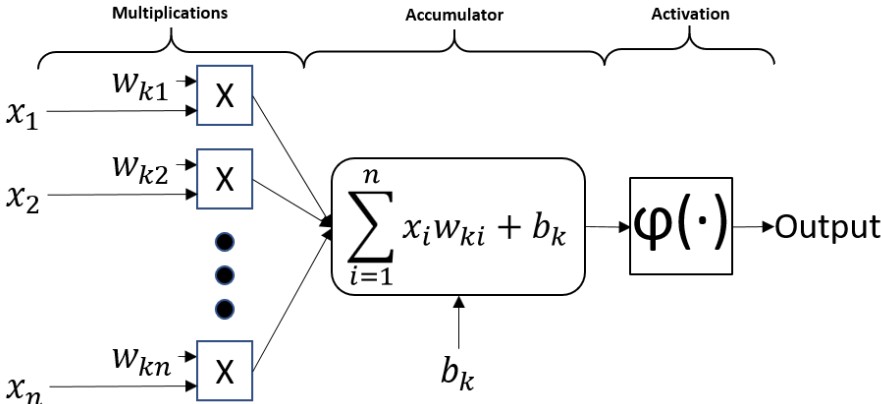

**Figure 3.** Structure of a neuron.

A type of frequently used NN is constructed by arranging neurons in layers where all the neurons in every layer connect to each neuron in the adjacent forward layer, i.e., fully connected FFNN. This type of network is illustrated in Figure 4.

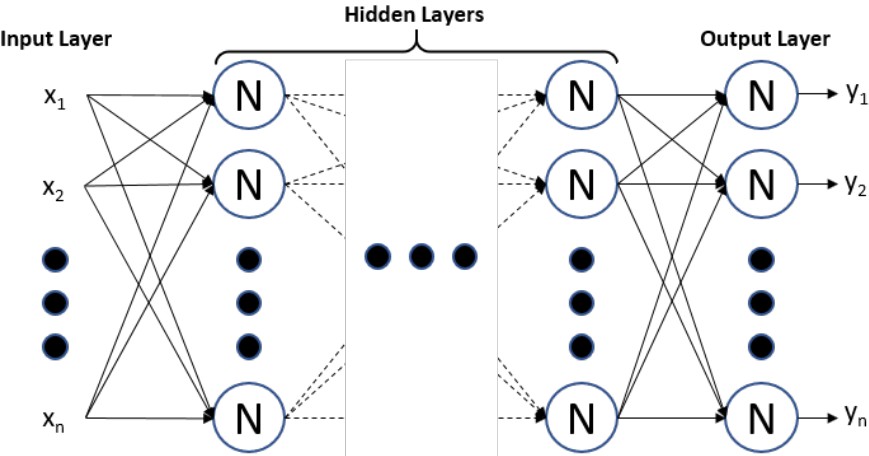

**Figure 4.** General structure of a feed-forward neural network.

By using the NNs structure, it is possible to derive the number of fundamental operations for each layer depending on its input and output count. Let $N_{in}$, $N_{out}$, $N_{add}$, $N_{mul}$, $N_{act}$ be the number of inputs, outputs, adders, multipliers and activation function operations, respectively. The number of

multipliers and adders corresponding to a fully pipelined implementation for a neuron is shown in Equation (1) and for each layer in Equation (2), respectively.

$$N_{mul} = N_{add} = N_{in} \tag{1}$$

$$N_{lmul} = N_{ladd} = N_{in} \times N_{out} \tag{2}$$

Every neuron has one activation function, thus for a layer, the theoretical number of required activation function calculations amounts to the number of outputs.

## 4. Consideration for the Design

Due to FFNN's parallel nature, a fully pipelined implementation can require more resources than it is available on the chip. Additionally, computational pipeline's throughput is determined by the slowest stage in the pipeline, thus provided with a limited communications bandwidth, a fully pipelined implementation can even be wasteful. Likewise, there is a risk of over-committing resources for some stages of the pipeline, while others are not suitable for pipelining at all.

For the implementation of an efficient pipeline, there is a need to develop a latency model of the NN. The finest timing (delay) unit we adopt is the clock period. Conventionally, an FFNN can be abstracted as a combination of neurons. If we denote the delay with $\tau$ and take into account data dependencies, as illustrated in Figure 3, neuron's delay can be characterized by):

$$\tau_{total} = \tau_{mul} + \tau_{add} + \tau_{act} \tag{3}$$

where $\tau_{mul}$, $\tau_{add}$ and $\tau_{act}$ are the respective delays of the total multiplication, addition and activation function operations.

By studying NN's structure and assuming every operation is characterized by some operation-specific constant delay of $\tau_c$, we can derive simple delay-resource relationships. For example, if all multiplications are done in parallel the multiplication delay model is:

$$\tau_{mul} = \left\lceil \frac{N_{in}}{N_{mul}} \right\rceil \times \tau_c. \tag{4}$$

As for the neuron's summation model, all the inputs of a neuron are summed up together (bias is treated as one of the inputs), which results in potential inter-dependencies for adders. Furthermore, depending on the clock frequency, chosen data type and routing, multiple distinct adder chains can be joined for processing in a single clock cycle. Nevertheless, for the convenience, we adopt a simpler addition-per-period model. Another important consideration is the complexity of adder scheduling, as different resource sharing arrangements can lead to different delay characterization. This is illustrated in Figure 5. Notably, many FPGAs employ a multiply-accumulate circuit in their DSP blocks; nevertheless, their use restrains the pipelining characteristics of the overall design.

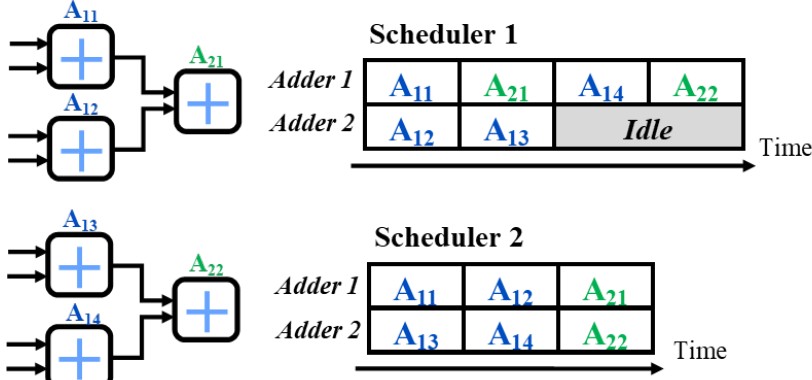

**Figure 5.** An illustration of different resource sharing policies, when two adders are shared between two neurons with a total of 6 needed additions. The first scheduler prioritizes the first neuron which results in an additional delay due to dependencies. The second approach prioritizes all of the neurons equally, leading to a more efficient resource sharing policy.

Previous considerations imply that the minimum delay for the summation phase can be achieved by using a tree-like adder structure which is:

$$\tau_{add} = \lceil log_2(N_{in}) \rceil \times \tau_c. \tag{5}$$

In general, if the adder count is limited, the addition layer's delay can be determined with the following simple Algorithm 1:

---
**Algorithm 1** Addition delay for single neuron

---
1: $N = N_{in}$
2: $\tau = 0$
3: WHILE $N > 1$
4:     $N = N - min(N/2, N_{add})$
5:     $\tau + +$

---

Although neuron is an intuitive abstraction, it subdivides architecture and respective delay models as its shown in Figure 6a. This approach omits resource sharing between parallel neurons. Therefore we propose a different abstraction shown in Figure 6b. In this approach, the neural network structure is separated into elementary layers, where each layer can be characterized by a specific resource-adder, multiplier or activation function.

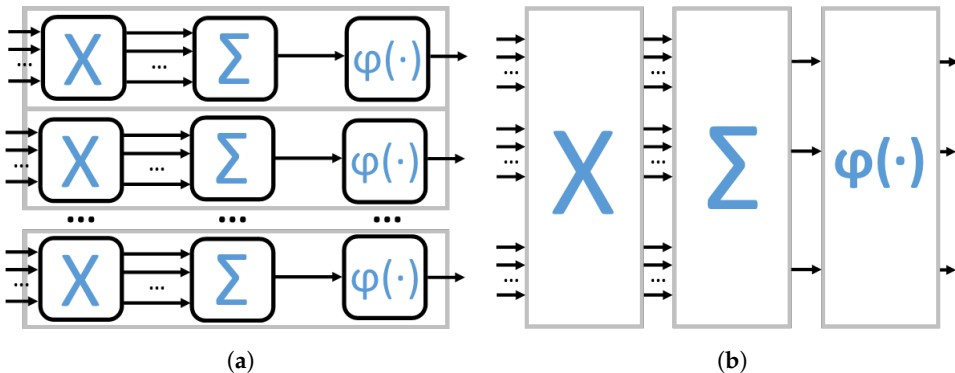

(a)                                                      (b)

**Figure 6.** Different delay models. (**a**) Delays are analyzed in terms of neurons in the layers (**b**) Joint delay analysis for "primitive" multiplication, addition and activation layers. (**a**) Conventional delay model for a single layer. (**b**) Proposed delay model with resource sharing for a single layer.

If the main bottleneck (either bandwidth limitation or some resource availability) is identified, it is possible to optimize the network by allocating the appropriate amount of resources for each layer. One of the most common limitations in an FFNN is the activation function as it is often challenging to pipeline.

The delay calculation procedure is provided in Algorithm 2. It takes into account that unused resources in one stage can be carried out to the next one. Variables are designated as follows: $N_{lin}-$ elementary layer's input count, $N_{add}-$ adder count, $N_{uadd}-$ adders used in current stage, $N_{cadd}-$ adders carried to the next stage, $\tau-$ delay cycles.

---

**Algorithm 2** Delay of elementary addition layer

---

1:　$N = N_{lin}$
2:　$\tau = 0$
3:　$N_{uadd} = N_{add}$
4:　WHILE $N > 1$
5:　　　　$N_{cadd} = N - N_{uadd}$
6:　　　　$N_{uadd} = min(N/2, N_{add} + N_{cadd})$
7:　　　　$N = N - N_{uadd}$
8:　　　　$\tau + +$

---

An important consideration for the ANN implementation is the choice of the data type. The floating-point data type contributes a high precision and range but is costly to implement and pipeline. Furthermore, recent studies [12,35] show that the fixed-point data types can be used with a small loss in precision, especially if anticipated by the ANN training procedure. Another important aspect is the possibility of using multiple data types within a single network. This can enable the implementation of larger networks, for example, by using data type of reduced precision for bigger layers of the network. Of course, the precision loss should be evaluated beforehand.

Another trade-off concerns the storage of the network coefficients. Coefficients can be stored in registers, thus ensuring parallel access to all of the coefficients, or written to on-chip memory which limits their accessibility. This can be decided depending on the performance requirements of the pipeline.

When examining the characteristics of the FFNN computation pipeline, the aforementioned considerations play an important role in achieving a balance between throughput, resource use and precision. These considerations lay the foundation for the throughput-optimized FFNN implementation described in the next section.

## 5. The Proposed Approach

A tool has been designed to automate the implementation of different networks, which takes FFNN topology as an input and generates C++ code for Xilinx HLS. The tool is based on the following methodology. First, the network topology is split into "elementary" layers, where each layer can be characterized by a specific resource, i.e., adder, multiplier, activation function, normalization function. Resource sharing is accomplished with multiplexing logic as shown in Figure 7. The implementation of this logic is managed by the HLS tool. Second, the network delay model is derived. This information enables the latency calculation for each elementary layer and determines the made choices regarding the FFNN's pipeline. Each layer's delay decreases by increasing the number of its allocated resources. In an optimal pipeline, all layers should have the same delay. This principle persists throughout throughput optimization calculations. The developed tool can be integrated into a larger workflow, e.g., as a part of CNN's classifier design. This is illustrated in Figure 8.

Besides the NN topology, the developed tool is supplied with a target interface for the FFNN accelerator IP core, which is either streaming or memory-mapped. Furthermore, the tool can be used in interactive mode, where the necessary configuration can be determined on-the-fly. Additionally,

the tool supports data type specification per elementary layer. Feedback from the HLS tools, e.g., timing estimation and resource use, is used iteratively by the user to find an optimal design.

One of the tool's main parameters which steer the code generation process is the maximum acceptable delay (expressed in clock cycles) for each stage of the pipeline. This parameter enables allocating "just enough" resources to comply with the given constraint-pipeline's initiation interval. An example of such a resource relationship is shown in Figure 9, where the same FFNN topology is fed into the developed tool while targeting a range of initiation intervals. Furthermore, the developed tool supports different normalization, addition, multiplication and activation layers and can easily extend to incorporate new calculations or operations described with LUTs. For example, activation function can be calculated analytically by using Xilinx and LOGICore IPs or by implementing it in a LUT. Nevertheless, when using Xilinx IP cores, the user should check the core's support for the targeted data type.

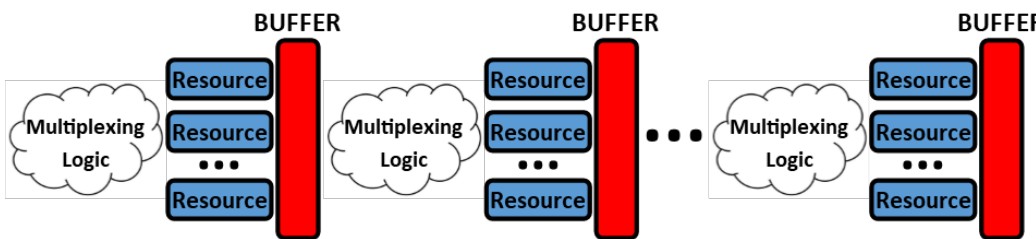

**Figure 7.** Proposed resource implementation scheme.

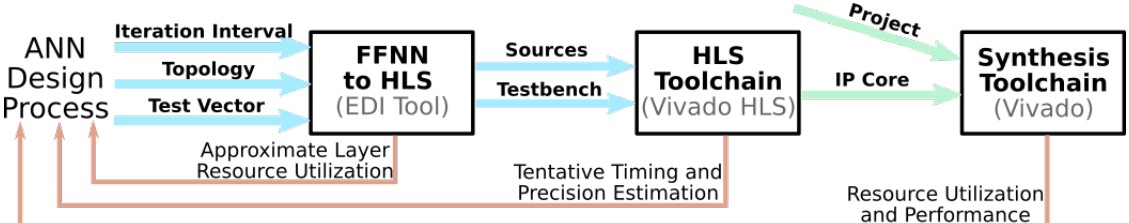

**Figure 8.** Tool flow of the proposed ANN development.

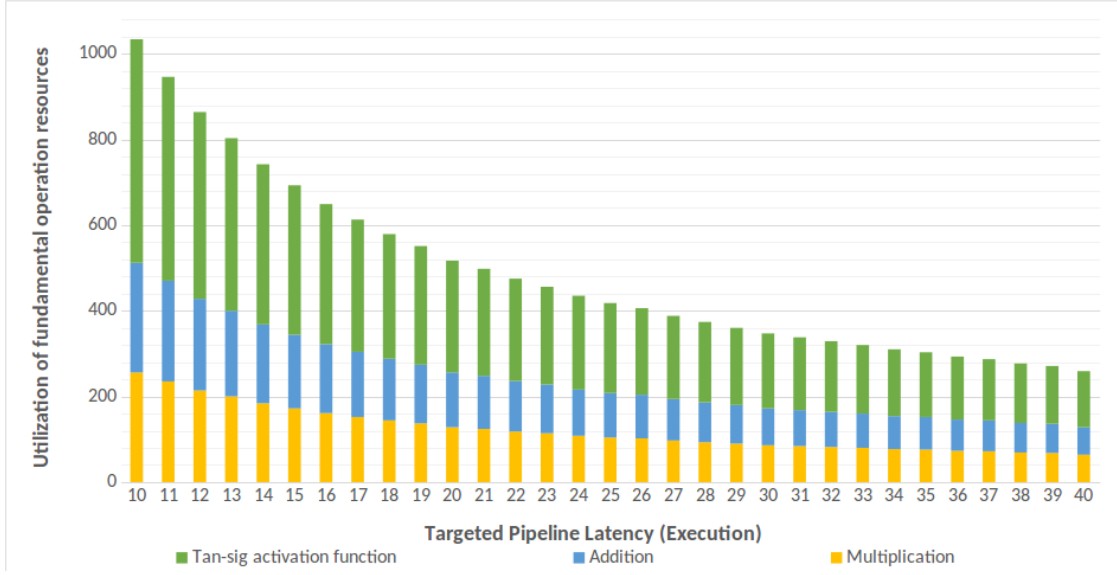

**Figure 9.** "Elementary layer" resource dependence on the targeted pipelining latency for a 17-40-30-20-4 FFNN topology [31]. The normalization layers are not displayed because their usage varies minimally (1–2 resources).

The generated code consists of NN implementation files in C++ programming language and data files for the NN coefficients. Every elementary layer is abstracted as a function. Source code incorporates *pragma* directives for the HLS compiler, which guide the synthesis of the RTL description. The developed tool is open-source and made available online (http://git.edi.lv/rihards.novickis/generation_tool_hls_c_fully_connected_feed_forward_neural_network) under MIT license. The generated FFNN IP cores are tested using two hardware interfaces: Memory-Mapped (MM) interface for the latency estimation and Streaming (ST) interface for latency and throughput estimation. MM interface implies active control from the Micro Processor Unit's (MPU's) side, but ST interface is set up by using LogiCORE DMA IP core [36] and AXI High-performance interface [37], thus transactions are carried out with a non-cached memory. In release implementation, memory coherence can be achieved by using coherent Processing System (PS)-Programmable Logic (PL) communication interfaces, by using non-cached memory [38] or actively managing caches by flushing or other mechanisms.

All tests were performed on Xilinx Zynq ZC702 SoC evaluation board using bare metal stack with FPGA logic being clocked at 100'MHz frequency. Timing measurements are made by using the Cortex-A9 Snoop Control Unit's (SCU's) timer. MM-based accelerator variant of the core is benchmarked by running IP core for 100 times and measuring its time of operation. Timer measurements are made before launching IP and after receiving the interrupt signal indicating the end of the core's operation. It must be noted that there are sub-microsecond measurement errors brought in by the internal interconnect structure. The ST interface was tested 100 times with 10,000 continuous neural network data points for estimating throughput and 100 times with just a single data point for estimating latency.

Activation functions for the fixed point networks were implemented using LUTs with a 12-bit input where 4 bits are allocated for the integer part of the number, while the LUT's output type coincides with the globally set network's data type. The related published work omits the trained coefficients and even the training data, therefore, all weights and biases were generated randomly in the range of the used data type.

Notably, the solution presented in this article is designed for FPGA implementation and does not support on-the-fly reconfiguration for different network topologies as opposed to some other solutions. Nevertheless, the developed tool and methodology can be integrated into larger ANN training and deployment procedures, thus omitting the need for a generic architecture, of course, if an instant or continuous reconfiguration is not a requirement.

## 6. Comparison with Other Approaches

As mentioned previously, the majority of the related work does not provide the actual coefficients of the FFNNs therefore, for these networks, we chose to evaluate the synthesized circuits by comparing their performance and use of the digital resources.

FFNN resource use, implementation details and comparison with related work [23,24,26,27] are illustrated in Table 2. "Target Initiation Interval" denotes the targeted number of cycles for each pipeline stage which is fed into FFNN generation tool for throughput optimization. "Theoretical Initiation Interval" and "Theoretical Latency" illustrates generated IP core's theoretical performance obtained from Xilinx Vivado HLS tool. $\sigma$ denotes the standard deviation of the measurements. Latency is provided in clock cycles (synthesized for 100'MHz clock). A detailed comparison with [28] is shown in Table 3 as the author targets the same FPGA technology and provides detailed data on the FPGA resource use. The A and B refer to resource-saving implementations while C and D use parallel neuron calculation within each layer. Additionally, A and C variants employ a single pair of the floating-point multiplier-adder blocks, whereas the remaining variants use two pairs.

An interesting finding in Table 2 is the absence of DSP blocks in some implementations. This can be explained by the use of low precision data types, i.e., because coefficients are "hardcoded" into the fabric, synthesis tools are able to perform optimization.

**Table 2.** Implementation resource use and benchmark result comparison table with [23,24,26,27].

| Topology | Interface | LUTs | FFs | DSPs | BRAM | Target Initiation Interval | Theoretical Initiation Interval | Theoretical Latency | Latency | | Throughput (Samples/s) | |
|---|---|---|---|---|---|---|---|---|---|---|---|---|
| | | | | | | | | | Original | Achieved | Original | Achieved |
| [23] 2-2-1 | Memory-mapped | 246 (0.46%) | 118 (0.11%) | 6 (2.73%) | 3 (2.14%) | 1 | 2 | 6 | 34 ns | 555 ns $\sigma = 3.4$ ns | 29,412,000 | 1,803,200 |
| | Streaming | 205 (0.39%) | 135 (0.13%) | 6 (2.73%) | 3 (2.14%) | 1 | 2 | 9 | | 2.68 µs $\sigma = 37.5$ ns | | 49,272,000 |
| [24] 2-4-1 | Memory-mapped | 980 (1.84%) | 800 (0.75%) | 48 (21.82%) | 9 (6.43%) | 1 | 2 | 10 | 44 ns | 586 ns $\sigma = 3.4$ ns | 2,272,700 | 1,705,600 |
| | Streaming | 983 (1.85%) | 946 (2.92%) | 48 (21.82%) | 9 (6.43%) | 1 | 2 | 12 | | 2.69 µs $\sigma = 44$ ns | | 49,231,000 |
| [26] 4-8-3-3 | Memory-mapped | 1304 (2.45%) | 912 (0.86%) | 0 (0.0%) | 3.5 (2.5%) | 1 | 4 | 12 | 1.16 µs | 620 ns $\sigma = 8.7$ ns | 862,070 | 1,612,400 |
| | Streaming | 1356 (2.55%) | 1028 (0.97%) | 0 (0.0%) | 3.5 (2.5%) | 1 | 4 | 19 | | 2.78 µs $\sigma = 91$ ns | | 24,796,000 |
| [27] 1-5-1 | Memory-mapped | 257 (0.5%) | 78 (0.1%) | 0 (0.0%) | 1.5 (1.1%) | 1 | 3 | 5 | 683 ns | 578 ns $\sigma = 9.5$ ns | 1,463,100 | 1,730,100 |
| | Streaming | 257 (0.5%) | 81 (0.1%) | 0 (0.0%) | 1.5 (1.1%) | 1 | 3 | 7 | | 2.68 µs $\sigma = 35$ ns | | 33,005,000 |

**Table 3.** Implementation resource use and benchmark result comparison table with [28] (Theoretical II = 100, Theoretical Latency = 987). (LUT-Look-up-Table, FF-Flip Flop, DSP-Digital Signal Processor core, BRAM-Block Random Access Memory).

| Implementation | LUTs | FFs | DSPs | BRAM | Latency | Throughput (Samples/s) |
|---|---|---|---|---|---|---|
| [28]A | 2232 (4.2%) | 1210 (1.1%) | 2 (0.9%) | - | 33.1 ms | 30.2 |
| [28]B | 3306 (6.2%) | 1326 (1.3%) | 4 (1.8%) | - | 24.7 ms | 40.5 |
| [28]C | 41,297 (77.6%) | 33,395 (31.4%) | 33 (15.0%) | - | 5.7 ms | 175.4 |
| [28]D | 51,028 (95.9%) | 35,655 (33.5%) | 65 (29.5%) | - | 3.5 ms | 285.7 |
| Impl. Memory-Mapped | 30,197 (56.8%) | 55,231 (51.9%) | 122 (55.5%) | 6 (4.3%) | 10.4 µs $\sigma = 0.011$ | 96,435 |
| Impl. Streaming | 31,246 (58.7%) | 56,067 (52.7%) | 122 (55.5%) | 6 (4.3%) | 12.5 µs $\sigma = 0.027$ | 997,852 |

Although the precision of the related work's implementations cannot be compared as the estimated coefficients and training procedures are not provided, recent studies suggest that it is possible to acquire good results by using a low precision data type [12,35]. This can be achieved by adopting an additional training stage where the targeted data type is used, as opposed to a standard training procedure which uses floating-point data type.

Results in Table 2 illustrate that ST-based implementation throughput approaches the theoretical Initiation Interval and can be characterized with about 2.7 µs latency. This is a result of different factors, e.g., DRAM controller delay, interconnect hierarchy and high-performance interface buffering logic. ST interface implementations provide higher throughput than any of the related work implementations, which is due to the pipelined nature of the accelerator, thus making it suitable, as an example, for the realization of virtual sensors.

Theoretical and practical measurements for the MM-based implementations are slower than implementations presented in [23,24], although [23] does not provide a practical implementation. Of course, additional latency, in this case as well, is introduced by the interconnect hierarchy. Nevertheless, the presented approach performs better than [27]. The results in Table 2 suggest that the developed approach excels at maximizing NN throughput. This is advantageous for virtual sensor use-cases, one of which is further examined in Section 7.

Although for the topology presented in [28] the hyperbolic tangent function was implemented using Xilinx LOGICore IPs, our implemented solution outperforms any of the versions presented in the paper. This suggests the suitability of the available IP cores for a floating-point implementation. The reason for such an impressive performance difference is the distinct design goals. In Reference [28], the author prioritizes on-the-fly reconfiguration of the network, while we target maximum throughput of the network.

## 7. Virtual Sensor Use-Case

To validate the developed methodology, a previously published [31] use-case has been chosen. The publication brings forward an interesting torque-vectoring application where a controller is proposed to estimate realistically unmeasurable signals of the vehicle's future dynamics. These signal values depend on the different control actions, therefore, it is necessary to process as many data samples as possible. For more information on the algorithm, the reader is redirected to [31]. This work provides not only a realistic use-case for the throughput-optimized implementation of the FFNN, but the coefficients and data sets are available as well.

For approach validation, an 8-16-12-8-4 topology was selected. The particular trained NN aims to estimate vertical forces on all four wheels of the electric car. The network is fed with the following inputs: steering angle, longitudinal acceleration, vertical acceleration, roll rate, pitch rate, yaw rate and lateral acceleration derivative. The inputs of the network are generated using automotive simulator Dynacar (http://www.dynacar.es/en/home.php), which runs a model of an actual vehicle.

The network topology was fed into the designed tool targeting Streaming (ST) interface, different iteration intervals and data types. The choice of ST-interface was motivated by the requirement of the virtual sensor, i.e., the necessity to maximize the throughput of estimations. The produced code was further processed by the HLS toolset, and the resulting IP core was exported into the synthesis toolchain. Latency for each of the cores was measured by using (a) integrated logic analyzer (realized using FPGA logic) and (b) bare-metal software. As expected, integrated logic analyzer measurements are equivalent to the ones predicted by the HLS tools, while the software measurements provide a realistic baseline as it incorporates delays brought in by DMA engines, SDRAM controller and SoC's internal routing logic.

The implementation results are provided in Table 4. The output values of the network are proportional to the vertical forces on the wheels (output normalization has been omitted). Standard deviation for the throughput measurements refers to the time of the DMA transactions. Sigmoid activation function has been implemented with LUTs, where the input corresponds to the NN type while the output is specified by the table.

**Table 4.** Configuration settings, FPGA resource use and performance metrics for 8-16-12-8-4 FFNN [31].

| NN Data Type | Act. Func. | Iter. Interv. | Resource Utilization | | | | | | | | Latency (μs) | Throughput (Samples/s) | Absolute Mean Error |
| | | | BRAM | | DSP | | Registers | | LUTs | | | | |
| | | | Tot. | % | Tot. | % | Tot. | % | Tot. | % | | | |
| Fixed 14,5 | LUT 8,1 | 32 | 8 | 20.79% | 14 | 5.71% | 22,452 | 21.1% | 8958 | 16.84% | 10.221 $\sigma = 0.03$ μs | 1.52 M $\sigma = 0.046$ μs | 0.0232 $\sigma = 0.014$ |
| Fixed 14,5 | LUT 8,1 | 24 | 8 | 20.79% | 20 | 9.09% | 22,123 | 20.79% | 8881 | 16.69% | 8.067 $\sigma = 0.01$ μs | 2.02 M $\sigma = 0.063$ μs | |
| Fixed 14,5 | LUT 10,1 | 24 | 10 | 21.02% | 20 | 9.09% | 22,369 | 21.02% | 9607 | 18.06% | 8.058 $\sigma = 0.111$ μs | 2.02 M $\sigma = 0.063$ μs | 0.0189 $\sigma = 0.011$ |

Some of the configurations achieve an impressive two mega-samples per second, which even overcomes the FPGA performance results in the original article [31]. The approach potentially could outperform GPU implementation with six mega-samples if a bigger FPGA chip is used while having the benefit of lower power consumption. As expected, the reduction of the targeted iteration interval improves performance but increases resource use as well. Notably, the improvement in the precision of the LUT output improves the overall precision of the network.

## 8. Conclusions

This work presents a novel throughput and latency aware FPGA implementation methodology for fully connected FFNNs. In this method, the NN is split into elementary layers, e.g., adders, multipliers, activation functions, where each layer has its delay characterization. This model steers the generation of high-level C++ programming language description of the topology, which is suitable for HLS tools and IP core generation. The developed workflow is designed for optimal throughput and can be used as a part of a larger ANN implementation workflow, e.g., classifier implementation for CNNs. The proposed workflow was evaluated with a virtual sensor implementation as it can provide a high sampling rate when compared to other implementations.

HLS tools can provide an advantage in terms of algorithm implementation and verification due to the usage of high-level languages, thus omitting the cycle-by-cycle behavioural description of the hardware. This article demonstrates that the proposed implementation paradigm together with high-level workflow can outperform conventional FFNN implementation approaches. Notably, the proposed approach is not generic, i.e., tool flow has to be repeated for every topology and NN

training procedure. Nevertheless, the developed workflow can be integrated into a larger DNN system development pipeline.

The proposed approach was compared with NN implementations in the related work. Although the latency characteristics of the generated accelerator IP core can be worse than in other design approaches, the developed tool's aim is the design of optimized processing pipeline, which in turn yields superior throughput when compared to the other approaches. This suggests the suitability of the developed approach in emerging use-cases, for example, in virtual sensor design.

The virtual sensor use-case was picked up from work done by Dendaluce et al. [31], where FFNN is used for electric vehicle's normal force estimation. This choice was motivated by the availability of the trained model. Different optimization and data type variants were implemented, and the resulting implementation even succeeds the one presented in the original work.

The developed tool for converting FFNN topology to the description for HLS is made available online under MIT license. The proposed approach still could be improved because it failed to achieve one cycle iteration interval for smaller FFNN topologies where it is certainly possible. Additionally, the use of HLS introduces additional logic, which can be a limitation for the implementation of more complex networks. Future work involves adding the support for ML compiler's-Glow-low-level intermediate representation language [39], supporting additional activation functions, improving the delay model and enabling support for modern recursive neural networks.

**Author Contributions:** Conceptualization, R.N.; Funding acquisition, K.O.; Methodology, R.N. and D.J.J.; Project administration, K.O. and M.G.; Software, R.N. and D.J.J.; Supervision, K.O. and M.G.; Validation, R.N. and D.J.J.; Visualization, R.N.; Writing—original draft, R.N. and D.J.J.; Writing—review and editing, K.O. and M.G. All authors have read and agreed to the published version of the manuscript.

**Funding:** This work is the result of activities within the 3Ccar project, which has received funding from ECSEL Joint Undertaking under grant agreement No. 662192. This Joint Undertaking received support from the European Union's Horizon 2020 research and innovation program and Germany, Austria, Czech Republic, Romania, Belgium, United Kingdom, France, Netherlands, Latvia, Finland, Spain, Italy, Lithuania.

**Conflicts of Interest:** The authors declare no conflict of interest.

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
