# Peer review of "An Approach of Feed-Forward Neural Network Throughput-Optimized Implementation in FPGA"

_electronics, doi:10.3390/electronics9122193_

Round 1
Reviewer 1 Report
This work presents an approach for fully connected feed-forward neural networks (FNNNs) with the development of an FPGA based accelerator. The article looks interesting apart for some important references that are missing. Kindly see below.
- Fine, Terrence L. Feedforward neural network methodology. Springer Science & Business Media, 2006
- Foumani, Seyedeh Niusha Alavi, Ce Guo, and Wayne Luk. "An FPGA Accelerated Method for Training Feed-forward Neural Networks Using Alternating Direction Method of Multipliers and LSMR." arXiv preprint arXiv:2009.02784(2020)
This paper has some grammatical and languages issues, that needs to be looked into. Kindly proofread the article again.
Reviewer 2 Report
The authors present a tool implement in FPGA devices feed-foward neural networks.
My main concerns are:
- Section 2,6 and 7: The authors based its comparison in paper [19-14] but some commercial approaches are commented in section 2. Whay these approaches are not included in section 6 and 7 for the sake of comparing results?
- Section 4. The cost of the resources involved in adderr sharing must be equivalent to using a fully parallel architecture when the depth of the adder trees is small. The author must elaborate on this point.
- Section 4. The authors are not considering that it is possible to tune the fixed-point format for each network stage.
- Section 5. It is not clear if there is an automated method to consider the Vivada timing estimations to refine the architecture of the network. The authors must explain thoroughly the flow of the presented tool.
- Section 6. I do not see any conclusions from Table 2.
- Section 6. The details of Table 3 are not explained. What are A, B, C, etc.?
Reviewer 3 Report
First and foremost, it is not clear at all how you achieve the stated goal, of equalizing the II of the various "layers".
Secondly, it is not clear why the layers would need to be adder and multiplier.
The elementary operation of a FFNN is a dot product, which involves a MAC. Spltting addition from multiplication requires some care, and ensuring EFFICIENT STREAMING COMMUNICATION between the layers is paramount, yet it is not discussed at all.
Moreover, most deep neural networks today are CONVOLUTIONAL. How does your approach extend to those?
How do you address quantization and quantized integer numerical representations?
Why do you focus on tanh, while most deep neural netorks today use RELU, which is much faster to implement and better for training (less vanishing gradient issues)?
Finally several recent METHODOLOGY papers, e.g.:
- Your ref 18,
- FP-DNN: An Automated Framework for Mapping
Deep Neural Networks onto FPGAs with RTL-HLS
Hybrid Templates, from the same group - FINN-R: An End-to-End Deep-Learning Framework for Fast
Exploration of Quantized Neural Networks, by Xilinx researchers
address the design of FFNNs with more generality and delay. How do you compare against them? How do you INETRFACE with FFNN architectural exploration tools like Tensorflow or Pytorch??
You seem to simply provide a library of parameterizable HLS modules for the various layers of an FFNN. Without more details on what exactly they are, what is their loop structure, how is storage managed, and how they are parameterized to geenrate different IIs and resource usages, I cannot evaluate the novelty of your work, nor understand what is your contribution.
Reviewer 4 Report
As stated already in previous version of the review (although this is formally a new submission), the submitted paper is interesting and also important. Both of these features are based on the fact that exploitation of neural networks nowadays forms significant and increasing base for applications in various fields, especially in image processing and signal processing, pattern recognition, etc. The paper is nicely written, this was also true in the previous version of the submission to my opinion, while some details of the paper still may benefit from changes. Specifically (ordered "in line" with the previous review to keep consistency):
- I believe that the section 5. The proposed appraoch (originally Implementation) has been renamed according to the recommendation, the this point is resolved OK.
- The "5. Implementation" section now does include much more details (the original point was regarding that "is did not") so the reproducibiloity of the results of the paper is not much better. Still, perhaps the source codes for FPGA etc., could be released (not a must but would be fine).
- The contributions of the paper are nicely summarized in the introductory section of the paper and they are in line with the actual contect of the paper. (The previous comment "complained" that the contribution was not "backed" by the text of the paper). Therefore, this point is resolved OK.
- Table 2 (addressed in the previous version of the review, sorry erroneously as Figure 2) is now better explained in the text. Anyhow, some better explanation directly in the Table 2 would be fine, e.g. streaming instead of ST, memory mapped instead of MM, Initiation Interval instead of II (that looks like "Roman number 2 by the way) would improve readability of the Table a lot. There abbreviations are explained in the text of the paper but it is quite difficult to look for them trying to understand. Please, improve Table 2.
- Figure 9 (previously Figure 8), that I criticised regarding its legend, is now improved and quite well readable. This point is resolved OK.
- Table 3 was criticiesd in the previous version of the paper for showing comparison of the achieved results to the state of the art. It is now understandable why the achieved results outperform the state of the art.Anyhow, I would suggest not to use MM and ST (and instead use memory mapped or streaming) to improve readability of the table. I also believe that it would be useful to briefly summarize (somewhere close to Table 3) what is the source of so impressive difference in throughput.
I believe that addressing the above (though now quite minor) remarks would improve the paper and perhaps make it easier to read read.
Round 2
Reviewer 3 Report
You say that:
> Furthermore, adders and multipliers are one of the primary resources of an FPGA. By
> separating these arithmetic operations, we can have a better control over resource
> allocation.
No. Integer multiply-and-accumulate in general can be performed using simply a multiplier, thus saving significant resources and timing. On an FPGA, the basic resource for this computation is a DSP, which can perform one multiplication and one or two additions (pre-add and post-add) depending on the architecture.
> In our approach, we did not address CNNs as we are concentrating
on the most resource and compute-intensive fully connected layers and maximizing their
> throughput.
No. Convolutional layers occupy 85-90% of the total workload of a CNN.
> For the sake of clarity, the quantization is performed directly to the coefficients. The
> floating-point coefficients are rounded to the nearest fixed-point value of the respective
> layer. This is performed by the HLS tool and its library for number fixed-point
> representation.
No. Quantization is a phase of the design space exploration for FPGA implementation of DNNs that must carefully choose the data representation (range, scaling, offset). Once you have done that, the HLS tool can then synthesize the chosen representation. The quantization/error rate trade-off is one of the most fundamental trade-offs when implementimg DNNs in HW.
> While
> FINN-R and FP-DNN focus on extensibility and support for different workflows and
> platforms,
No. They focus on one flow, HLS, for one platform, Xilinx. And unless you can show advancement with respect to state of the art, your paper cannot be accepted. You must show that you achieve better results than competitors.
They have exactly the same gaol: implement DNNs in HW. And they do it (as far as I can tell) much better.
> Finally, we would like to clarify that, although the developed tool is an important
> contribution of our research, the aim is to report on the success of the developed
> methodology for FFNN throughput optimization, which, as shown in the article, provides
> an unprecedented potential for virtual sensor implementation.
Then you need to submit your paper to another journal, focusing on slip prevention. This will allow people from that specific application domain to judge how impressive is your claim of 2MS/s on an FPGA. I cannot.
And you need to change the title, from "an approach..." to "An efficient low-cost implementation of anti-slippage sensing on FPGAs".
Suggested journals:
- IET automotive electronics
- IEEE Trends in automotive electronics
- Springer International journal of automotive technology
Author Response
Please see the attachment.

This manuscript is a resubmission of an earlier submission. The following is a list of the peer review reports and author responses from that submission.
Round 1
Reviewer 1 Report
- The submitted paper is interesting and also important. Both of these features are based on the fact that exploitation of neural networks nowadays forms significant and increasing base for applications in various fields, especially in signal and image processing, recognition, etc. The paper is nicely written, still, some details of the paper need changes before the paper is published. Specifically:
- I believe that the section "5. Implementation" should rather be renamed to e.g. "5. Design and Implementation" as the majority of contribution is in this section and perhaps the original name suggests "just implementation" of earlier described approach, which is not the case.
- The "5. Implementation" section does not describe in enough detail what is the method behind optimization of the network. While the contribution of the paper is visible even without this description, reproducibility of the method "suffers a lot" so it would be most appropriate to include details here.
- The contributions of the paper are nicely summarized in the introductory section of the paper. However, they are not completely backed by the text of the paper and this needs improvement. Specifically: "novel view on the implementation challenge" - OK, "novel implementation methodology" - not quite demonstrated - the design goal and latency issues shown fine but the "methodology" is not really shown that well, "development of ANN implementation generation tool" - some results shown but the algorithms in the tool are not shown, while this is the critical part of the contribution, this would be needed, "comparison with previous implementations" - yes, but something seems strange DRAM vs BRAM may bring limits in size (see also below).
- Figure 2 is quite confusing - i) it is misplaced in the text (within section 4 instead of 5) but, much more importantly, the content, although commented in the text of the paper, is not really understandable. Specifically the columns "Latency Original" vs "Latency Achieved" and "Throughput Original" vs "Throughput Achieved" are not explained and their meaning (and values) do not seem appropriate. Please, explain and/or fix.
- In Figure 8, it is not quite clear why the "legend" contains 8 different (probably) phases of the pipeline while in the graph itself, it is merely only 3 of them. Perhaps simplification of the graph would help understandability.
- Table 3 shows some comparison of the achieved results to the state of the art. It is very unclear why the achieved results would outperform the state of the art by more than 2 orders of magnitude. I am tempted to think that this is due to completely different memory interface and possibly also due to limited size of the neural network in the proposed implementation. Please, comment and explain the differences and also limits of the proposed implementation.
I believe that addressing the above remarks would improve the paper and perhaps make it more usable and suitable for reading.
Reviewer 2 Report
The writing is in general good and clear to understand, but the definite article “the” and the indefinite article “a/an” are missing in many cases. Please correct it.
Figure 8 legend color mislabeled. There are 8 legend colors, but the figure only has 3 colors. Please clarify it.
The throughput (samples/s) and latency are used as the measurement to compare different FPGA implementations of the FFNN. These parameters depend on the operating frequency of the FPGA. However, to compare the implementation performance of the FPGA design, the maximum operating frequency should be reported. This can be obtained by calculating using the worst negative slack in the implementation report.
Table 3: For reference [23]A to B, bigger throughput corresponds to smaller latency; The Impl. ST has a bigger throughput than Impl. MM, why is the latency bigger instead of smaller? Please explain.
Reviewer 3 Report
I am sorry to say that I have found no relevant contributions in this paper. Implementing a Neuronal network is not complex, even if you do it in a hardware description language, my students do it in one of their assignments and they have almost no experience with FPGAs. In this case the authors have used the Xilinx HLS environment which simplifies the process. I do not see any contribution in that. It has been previously done so many times, and if the authors want to contribute in that topic, they need to present a novel idea which is not the case. Do you want to improve the performance? That is trivial if the network is small, as it can be fully pipelined generating one output per cycle. Especially if you are using reduced precision as the authors propose. It is as simple as including a register in the output of each adder and each multiplier. And what about the hyperbolic tangent activation function? Please, can you explain me why are you using that function? Most machine learning environments are using ReLU as activation function by default. And ReLU can be implemented with a 2:1 mux! If you are looking for an efficient HW using a hyperbolic tangent activation makes no sense.
Moreover, the paper mentions some ideas that afterwards are not properly explained, such as reusing adders in the designs. How is that done, and what is the cost of that reuse. The interconnection network needed for that can be more expensive that the adder!
The experimental results present some numbers of an implemented Neuronal Network without explaining anything about which is the actual implemented NEURONAL NETWORK. That is no useful at all.
Reviewer 4 Report
This paper describes a tool that automatically generates a C++ artificial neural network description, which is suitable for high-level synthesis.
The paper is interesting and suitable for this journal.
The authors describe their approach and compares their results with state of the art solutions. Moreover, they provided a link for the source code of their tool.
I suggest inserting further details on the activation functions. Which are currently supported in the tool? Concerning the possibility of using a Look up table or Xilinx and LOGICore IPs for calculating the activation functions, is there a method for establish which solution is the best for the target application?
Please, give further explanation of table 3, what do the A, B, C and D letters mean for ref. [23]?
On lines 197-198, authors wrote that “Latency is provided in clock cycles” but in table 3 the latency is reported in ms. Please clarify this point.
Please, describe the layers used for the chart in Figure 8.
Minor points and typos:
- Please, carefully check the use of the possessive case.
- On page 1, line 17, please insert “a” between “for” and “variety”.
- On page 1, line 20, please insert “a” between “for” and “wide”.
- On page 1, line 9, please insert a comma after “improved”.
- On page 1, line 31, please insert “the” between “to” and “creation”.
- On page 1, line 34, please insert a comma after “article”.
- On page 2, line 61, please change “compliment” with “complement”.
- On page 2, line 64, please change “latency and clock period is” with “latency and clock period are”.
- On page 2, line 74, please insert “This” before “Approach”.
- On page 3, line 87, please insert “The” before “Exponent”.
- On page 3, line 88, please insert “The” before “Approach”.
- On page 4, line 104, please change “with” with “by”.
- On page 7, line 158, please insert “The” before “Tool”.
- On page 7, line 173, please insert “The” before “Tool”.
- On page 8, line 201, please leave a blank between “16” and “GB”.